# Rapid evolution and biogeographic spread in a colorectal cancer

Joao M. Alves [1,2,3], Sonia Prado-López [1,2,3], José Manuel Cameselle-Teijeiro [4,5] & David Posada [1,2,3]*

How and when tumoral clones start spreading to surrounding and distant tissues is currently unclear. Here we leveraged a model-based evolutionary framework to investigate the demographic and biogeographic history of a colorectal cancer. Our analyses strongly support an early monoclonal metastatic colonization, followed by a rapid population expansion at both primary and secondary sites. Moreover, we infer a hematogenous metastatic spread under positive selection, plus the return of some tumoral cells from the liver back to the colon lymph nodes. This study illustrates how sophisticated techniques typical of organismal evolution can provide a detailed, quantitative picture of the complex tumoral dynamics over time and space.

[1] Department of Biochemistry, Genetics and Immunology, University of Vigo, Vigo, Spain. [2] Biomedical Research Center (CINBIO), University of Vigo, Vigo, Spain. [3] Galicia Sur Health Research Institute, Vigo, Spain. [4] Department of Pathology, Clinical University Hospital, Galician Healthcare Service (SERGAS), Santiago de Compostela, Compostela, Spain. [5] Medical Faculty, University of Santiago de Compostela, Santiago de Compostela, Compostela, Spain. *email: dposada@uvigo.es

Cancer has long been recognized as a somatic evolutionary process mainly driven by continuous Darwinian natural selection, in which cells compete for space and resources[1]. With the increasing availability of high-throughput genomic data, several studies have started to explore the evolutionary relationships of tumor clones in order to identify the key molecular changes driving cancer progression[2], to better understand the subclonal architecture of tumors[3,4], and to determine the origins of metastases[5]. While sophisticated inferential methods have been put forward that make use of sequencing data to investigate the timing and the patterns of geographical dispersal of organismal lineages[6,7], their application in cancer research has only recently started[8,9].

In metastatic colorectal cancer (mCRC) many aspects underlying the dissemination of cancer cells to tissues beyond primary lesions have been difficult to determine. Although earlier models of mCRC progression have proposed a sequential metastatic cascade, with cells from the primary tumor first escaping to local lymph nodes from where they seed distant tissues[10], conflicting evidence has recently emerged, as some genomic datasets seem to favor an independent origin of distant and lymph node metastases[5]. Here, to better understand the tempo and mode of diversification of the tumoral cells within the human body, we analyze multiregional sequencing data from a single patient with mCRC under a powerful Bayesian framework, typical of organismal phylogenetics, phylodynamics, and biogeography. We are able to identify a rapid and early metastatic spread and multiple migration events, where both primary tumor and metastases diversify in parallel. Our results provide an unusually detailed picture of the complex evolution of tumor cell populations within a single individual, identifying tumor demographics and colonization patterns within a defined timeframe.

## Results

**Spatial distribution of intratumor genomic heterogeneity.** We obtained whole-exome sequencing data from 18 different locations of a microsatellite-stable mCRC (Fig. 1a). After filtering out germline polymorphisms and single-nucleotide variants (SNVs) in non-diploid regions, we detected 475 somatic SNVs with high confidence (Supplementary Data 1). A principal component analysis (PCA) of their allele frequencies showed a clear distinction between primary tumor and metastatic samples (Fig. 1b). Concordantly, we found a significant correlation between genetic and physical distances among these two groups, but not within (Supplementary Fig. 1). We identified several clonal alterations in known CRC drivers[11], including two copy neutral loss of heterozygosity events in *APC* and *TP53*, plus a non-synonymous mutation in *KRAS* (Fig. 1c, d). Moreover, we also observed a clonal non-synonymous mutation in *MSLN*, a plasma membrane differentiation antigen which is emerging as an attractive target for cancer immunotherapy due to its potential involvement in the epithelial-to-mesenchymal transition, a cellular process thought to be required for metastatic dissemination[12].

**Tempo and demographics of metastatic dissemination.** We obtained a Bayesian estimate of the phylogeny, under a relaxed clock model with exponential growth, of the 21 tumor clones identified with CloneFinder[13] (Fig. 2a). All the metastatic lineages grouped together with high support, suggesting a monoclonal origin. The age of the tumor was estimated to be 6.94–6.45 years (95% Highest Posterior Density (HPD): 9.98/9.16–4.43/4.36) prior to clinical diagnosis (PCD). Also, the results imply an early origin of the metastatic ancestor, 4.20 years PCD (95% HPD: 6.30–2.46) (Supplementary Fig. 2), diverging within a short period of evolutionary time (posterior median divergence time =

2.58 years) from the ancestor of the tumor sample (tMRCA) (Fig. 2b). Despite the lack of a significant overall departure from neutrality across branches, evidence of positive selection (i.e., ratio of substitution rates at non-synonymous and synonymous sites (d$N$/d$S$) > 1) was found for four specific branches in the phylogeny, including the ancestral lineage that gave rise to all the metastatic clones, pointing out changes potentially relevant for the acquisition of metastatic capabilities (Fig. 2a). The most notable mutation in this branch was a non-synonymous mutation in *ANGPT4*, an angiogenic gene known to promote cancer progression in multiple cancer types[14,15].

Furthermore, the Bayesian skyline plot (Fig. 2c) suggests that the tumor underwent a very rapid demographic expansion coincident with the diversification of both primary tumor and metastatic clades, before eventually becoming stationary. Interestingly, the expansion of the metastatic clade seems to slightly precede the one associated with the primary tumor. The posterior median estimate of the population growth rate per generation was 0.014 (95% HPD: 0.006–0.03), implying an average population doubling time of 193 days.

**Biogeographic history of cancer progression.** The colonization history of this tumor appears to have been quite complex. A dispersal-extinction biogeographic analysis placed the origin of sampled lineages around the geographical center of the primary tumor (Fig. 3a), subsequently radiating outwards in multiple directions. Additionally, we inferred with high confidence that the ancestral metastatic clone experienced an early long-distance dispersal to the liver (Fig. 3b), followed by a proliferation towards the nearby hepatic lymph nodes before eventually spreading back to the colonic lymph nodes. The number of implied migrations and movements was surprisingly high (Fig. 3c). Importantly, a distance-dependent model was heavily favored over a distance-independent model (Fig. 3d), suggesting an overall negative correlation between geographical distance and dispersal ability of the tumoral clones at the whole-patient level.

## Discussion

Collectively, our analysis provide a detailed picture of the evolutionary history of this tumor. While we are not the first ones applying Bayesian phylogenetics for cancer dating[8,9,16,17], previous attempts used sample trees and absence/presence mutational profiles instead of clonal phylogenies and clonal sequences, and therefore are subject to potential biases[18,19]. Besides, the evolutionary framework presented here has several advantages over previous approaches. For example, it is based on Bayesian estimates obtained only after contrasting competing evolutionary and demographic models under a rigorous model selection framework. Also, our biogeographic approach allows for the presence of the same ancestral clone at more than one location, and is able to consider the spatial distance among samples, unlike the approach of El-Kebir et al.[19]. On the other hand, our analyses imply a series of assumptions. In particular, it presumes that the clonal genotypes were appropriately reconstructed. Indeed, clonal deconvolution remains a very hard problem[13], and we cannot rule out some degree of uncertainty in the precise combination of mutations assigned to any given clone. Nevertheless, we were reassured to some extent by the fact that comparable clonal genotypes were obtained when using a different deconvolution approach[20] (Supplementary Fig. 3). In addition, in our analyses we used a mutation rate, 4.6E-10, experimentally derived from hundreds of CRCs[16]. However, it has been recently proposed that mutation rates in mCRC may be higher and could also vary among patients by an order of magnitude (i.e., 10E-9–10E-8)[17]. Higher mutation rates would essentially imply faster evolution

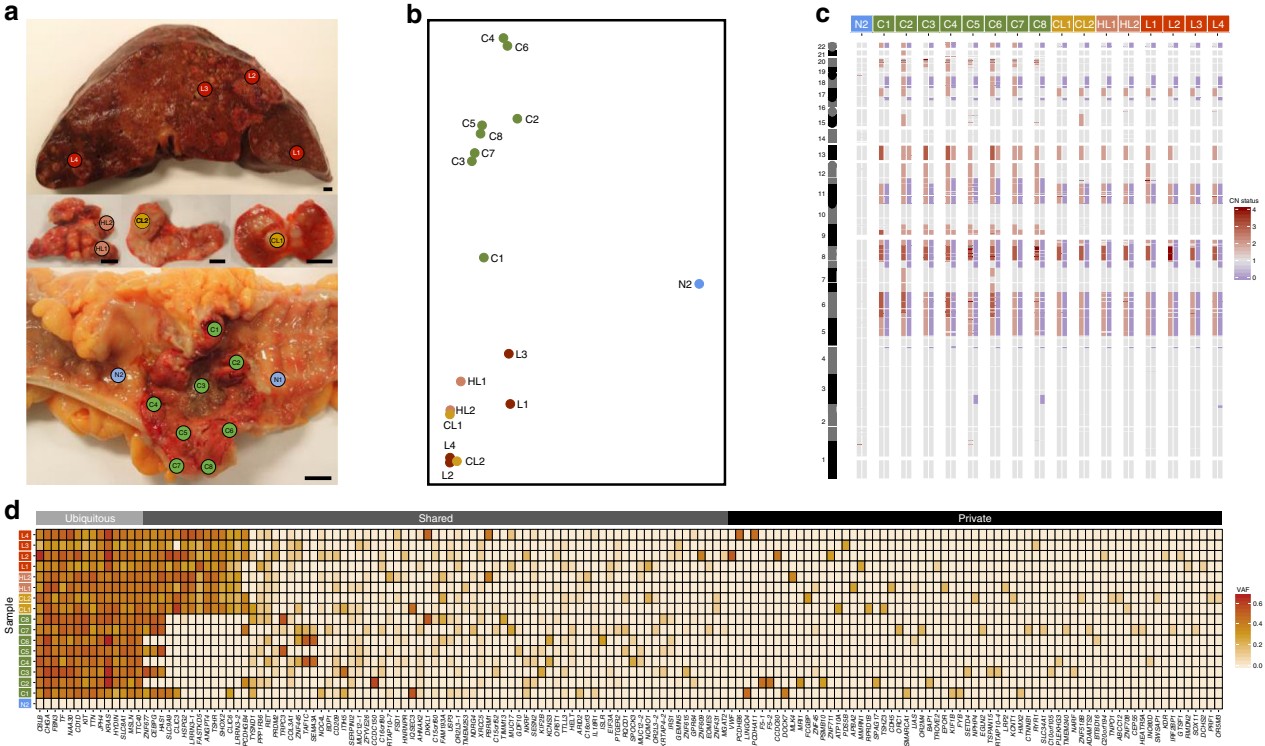

**Fig. 1** Genomic profiles of bulk tumor samples. **a** Multiregional sampling scheme. A total of 18 samples were collected, including two samples from healthy tissue (in blue), eight from the primary tumor (green), two from proximal colonic lymph nodes (gold), two from distal hepatic lymph nodes (salmon), and four from liver metastasis (red). Scale bar: 1 cm. **b** Principal component analysis (PCA) with variant allele frequencies (VAF) for all 475 somatic mutations detected. Each circle corresponds to a given sample, with colors highlighting the anatomical regions. **c** Heatmap depicting genome-wide allele-specific copy-number status (from 0 in blue to 4 in red) of healthy and tumor samples. Sample IDs are shown at the top. **d** Heatmap with the observed allele frequencies (from 0 in white to 0.65 in red) of somatic mutations identified in the sequenced samples. Here only the non-synonymous mutations are shown ($n = 156$), sorted according to their mean VAF across all tumor samples. Gene names are displayed at the bottom of the map. Each row represents a single sample

and shorter absolute divergence times. Nevertheless, it should be highlighted that regardless of the mutation rate prior used, the relative divergence time of the metastatic ancestor is not affected, indicating that an early divergence between mMRCA and tMRCA should still be supported. Moreover, our biogeographic model assumes that the geographical distances among samples more or less reflect the true migration likelihood of the tumoral clones. While we cannot prove that the distances used are realistic in this regard, different sets of distances resulted in similar biogeographic solutions (Supplementary Note 1, Supplementary Fig. 4).

Importantly, early metastases, such as the one described here, have already been proposed in mCRC[8,9,16,17]. Although Leung et al.[21] recently inferred a late-dissemination model in mCRC, they failed to provide quantitative measurements, and their timing of metastatic dissemination was simply determined by visual inspection of mutational trees, making their results difficult to interpret and compare with. Reinforcing the idea of an early cell dissemination, our results suggest a fairly rapid population increase during the parallel phylogenetic diversification of the metastatic and primary tumor clades. Although these analyses revealed a similar individual contribution of each clade to the overall variation in effective population size, the observed demographic trends are compatible with an early geographical expansion, and subsequent establishment of the metastatic lineages into new anatomical sites, together with the expansion of primary tumor populations to nearby areas.

Our biogeographic reconstruction revealed a pattern of metastatic dissemination in which the primary tumor directly seeded liver metastases without an apparent early involvement of the

lymphatic system. Previous studies have argued that metastatic spread in mCRC can potentially occur via the hepatic portal vein—a direct blood supply between the colon and the liver[5,22]. On this basis, metastatic dissemination in this patient seems to have started hematogenously, with a single episode of long-range dispersal across the hepatic portal vein into the liver, followed by a sequence of short-range migration episodes to nearby anatomical areas before eventually spreading to the colonic lymph nodes. While the latter colonization has not yet been described in mCRC patients, it might represent some type of self-seeding mechanism, as previously observed in mCRC in mice[23]. Interestingly, we observed a similar migration pattern, albeit less detailed, at the organ level using MACHINA[19] (Supplementary Note 2, Supplementary Fig. 5). In this case, all migration solutions suggested a single spread from the primary tumor towards the liver, followed by multiple migration events to the hepatic and colonic lymph nodes. Moreover, the MACHINA history with the smallest number of migration events also implied a parallel seeding of hepatic and colonic lymph node metastases from the liver.

In conclusion, we believe that our study demonstrates the utility of a sound evolutionary framework for exploring the spatio-temporal dynamics of cancer cell populations from multiregional sequencing data. By integrating concepts from population genetics, phylogenetics, and biogeography, we were able to resolve the spatial architecture of this cancer, connect phylogenetic events at time scales compatible with clinical observations, and recover past demographic changes shaping the spatial distribution of malignant clones. As more data continue to accumulate, future studies could extend these type of evolutionary

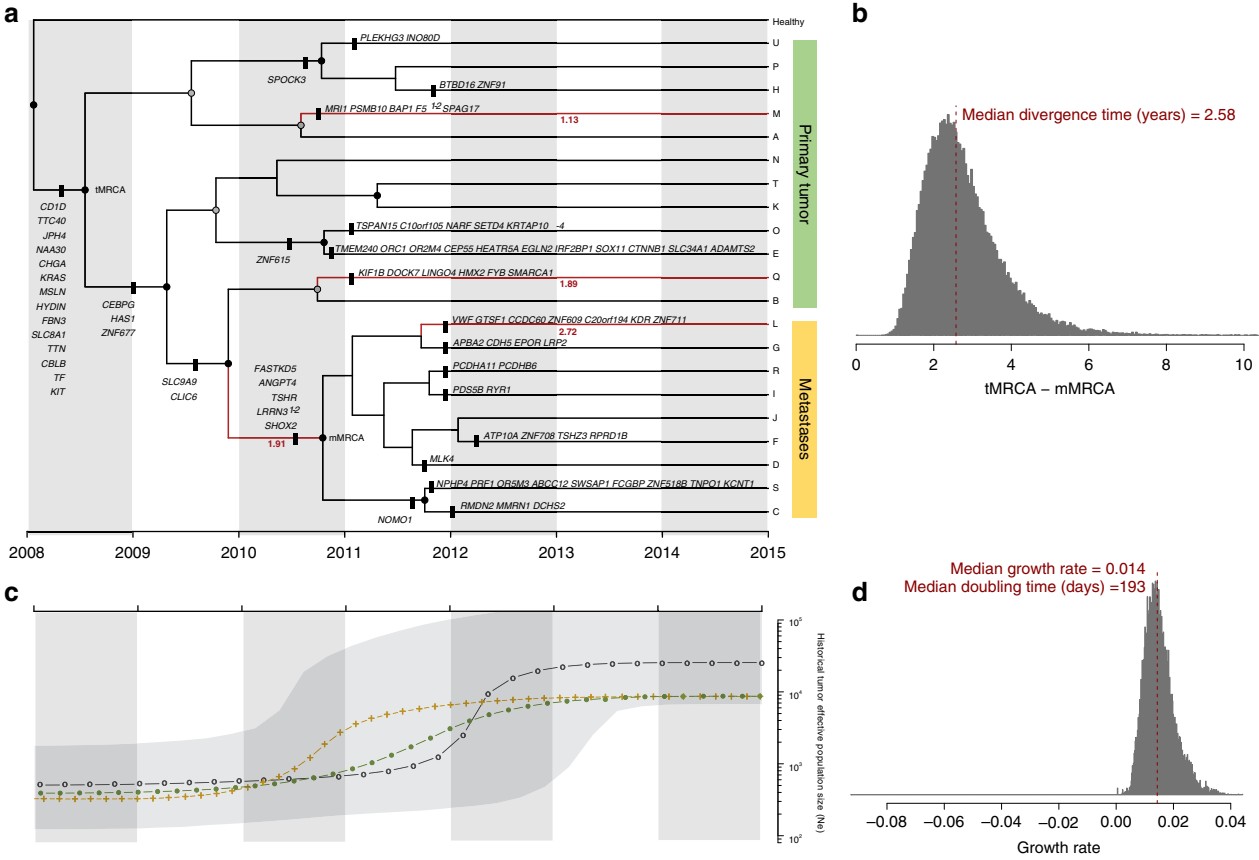

**Fig. 2** Phylogenetic and demographic reconstruction over time. **a** Maximum clade credibility (MCC) tree resulting from the BEAST analyses using the CloneFinder-derived clones. Tree nodes with posterior probability values >0.99 and >0.50 are indicated with black and gray solid circles, respectively. Clone IDs (A–U) are shown at the tips of the tree. The x-axis is scaled to years (assuming one generation every 4 days; see Methods). Only non-synonymous mutations are shown. Tree branches showing a dN/dS ratio >1 are highlighted in red together with the corresponding dN/dS value. **b** Posterior probability distribution of the relative divergence time in years of mMRCA in relation to the tMRCA (tMRCA minus mMRCA). The dashed red line depicts the median age estimate of the mMRCA. **c** Bayesian Skyline Plot analysis. The y-axis is in log scale. The black dotted line represents the historical effective population size of the entire cancer cell population (Ne). The gray shading illustrates the 95% HPD interval. Green and golden dotted lines correspond to the effective population sizes of the primary and metastatic populations, respectively. **d** Histogram illustrating the growth rate per generation of the tumor. The population doubling time is shown in days

analyses to other patients and cancer types, including polyclonal metastatic tumors[5], in order to obtain a more comprehensive and meaningful understanding of the cancer spread, which could ultimately be used to predict clinical outcomes and guide targeted treatments[24].

## Methods

**Sample collection**. A 51-year-old man was admitted to hospital and died shortly afterwards. The pathological assessment revealed a low-grade, moderately differentiated, adenocarcinoma of the descending colon, with multiple metastatic lymph nodes, liver metastases, a metastatic focus in the right diaphragmatic peritoneum, and multiple intravascular micrometastases in both lungs (pT4aN2bM1c)[25]. Immunohistochemical staining for four mismatch repair proteins (MLH1, MSH2, MSH6, and PMS2) confirmed that this tumor was microsatellite stable. During the warm autopsy, performed by J.M.C.-T., a total of 18 samples were collected, including 8 from the primary tumor (C1–C8), 2 from colonic lymph node metastases (CL1, CL2), 2 from hepatic lymph node metastases (HL1, HL2), 4 from liver metastases (L1–L4), and 2 healthy samples from the colon (N1, N2) (Fig. 1a). All samples included in this study were provided by the Biobank of I.D.I.S.-C.H.U. S. (PT13/0010/0068), integrated in the Spanish National Biobank Network, and processed following standard operating procedures with the appropriate approval of the Ethical and Scientific Committees (CAEI Galicia 2014/015). Written informed consent was provided by the patient's family.

**Tumor disaggregation and sorting**. Tumor samples and normal CRC tissues were frozen in liquid nitrogen, placed in dry ice, and transported to the laboratory. Next, samples were minced into pieces of 1 mm³ with a scalpel and digested by incubation in Accutase (LINUS) for 1 h at 37 °C. Thereafter, the cell suspension was

filtered with a 70 μm cell strainer (FALCON). The cell pellets were washed twice and suspended in ice-cold phosphate-buffered saline (PBS) and then stained for 30 min with the Anti-EpCAM (EBA1) antibody (BD). Following three successive washes in PBS buffer, flow cytometry analyses, and sorting of EpCAM-positive cells were performed with a FACSARIA III (BD Biosciences). Then, DRAQ5 and 7AAD dyes were added in order to select nucleated cells and exclude non-viable ones (Supplementary Fig. 6).

**DNA extraction and exome sequencing**. The DNA was extracted from the 18 samples using the QIAamp DNA Mini kit (QIAGEN), and whole-exome sequencing was carried out at 60× with the Ion Torrent PGM platform at the Fundación Pública Galega de Medicina Xenómica (FPGMX) at Santiago de Compostela, Spain.

**Detection of somatic variants**. Sequencing reads were aligned to the Genome Reference Consortium Human Build 37 (GRCh37) using the Torrent Mapping Alignment Program 5.0.7 (TMAP). After alignment, SNVs were called independently for all tumor and normal samples using a standalone version of the Torrent Variant Caller 5.6.0 (TVC). Following a similar approach to de Leng et al.[26], a set of high-stringency thresholds were used to retain high confidence bi-allelic calls, including a minimum coverage of 20× for both tumor and healthy samples, a minimum variant allele frequency (VAF) of 0.05, and a minimum nucleotide (Phred) quality score of 20. Given this rather stringent filtering strategy, sequencing errors are expected to be negligible. Germline polymorphisms were filtered by excluding variants present in the healthy samples. Copy-number profiles, as well as tumor purity estimates and global ploidy status, were obtained using the Sequenza toolkit[27] under default settings (binning window of 1 Mb).

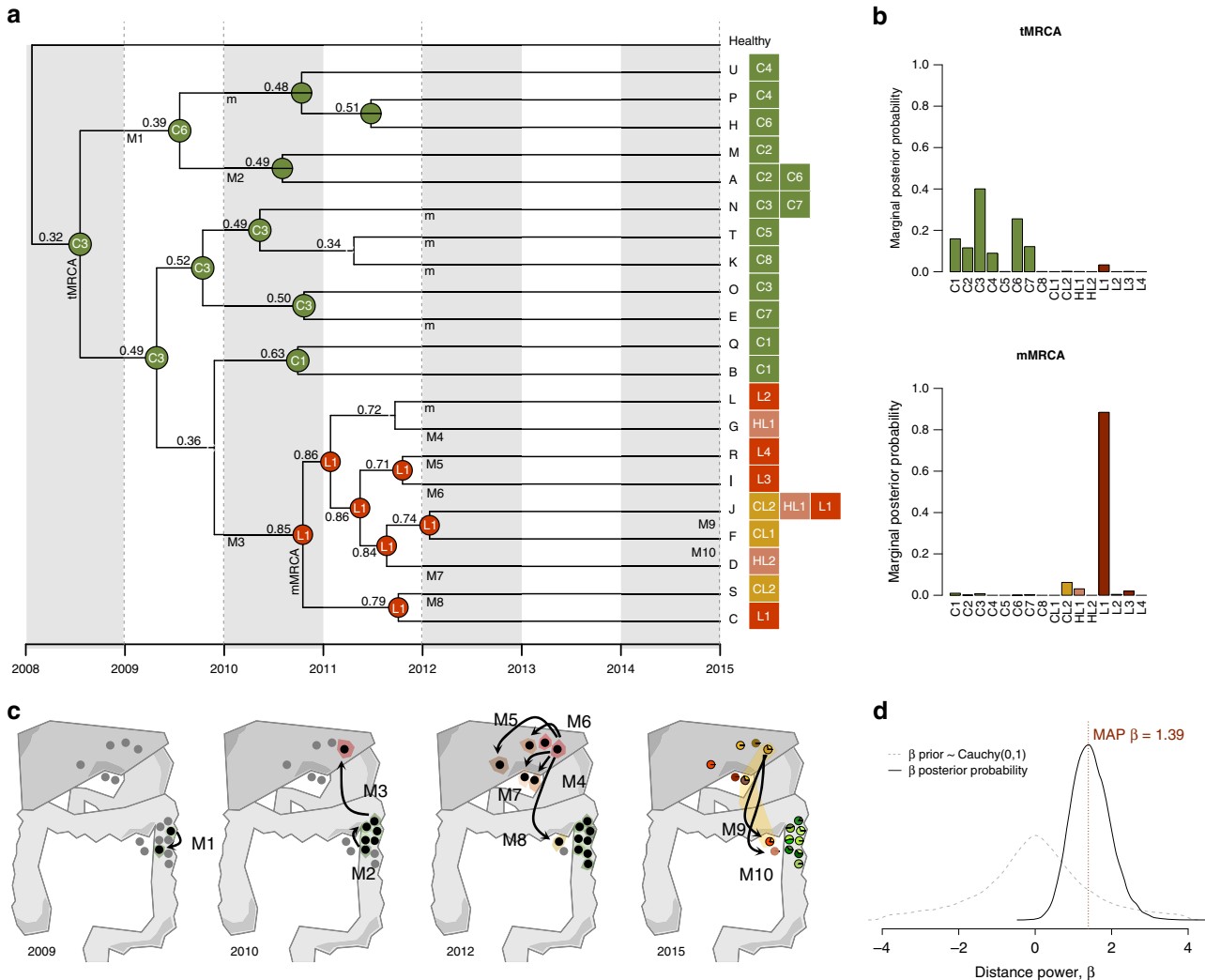

**Fig. 3** Inferred biogeographic history. **a** Biogeographic reconstruction from BayArea, describing the geographical range (i.e., the set of occupied locations) of the ancestral clones. At each tree node, the range with the highest posterior probability is depicted. The sample ID is shown for those ancestral nodes whose inferred area ranges are restricted to a single location. The locations where the extant clones (A–U) were sampled are shown next to the tips. Migration events are depicted in the panel below represented by an uppercase "M" and numbered (M1–M10). A lowercase "m" indicates the remaining migrations inferred. **b** Marginal posterior probabilities for the occupancy at single locations for the tumoral (tMRCA) and metastatic (mMRCA) ancestral clones. **c** Schematic representation of the clonal dynamics in anatomical space over four time points. From 2009 to 2012, samples where BayArea inferred the presence of tumor clones are highlighted in black. Colored areas surrounding samples anatomical location represent the inferred spatial distribution of the clonal populations. Arrows highlight the inferred migration events. **d** Comparison of the distance-dependent/independent dispersal models. The dashed gray line corresponds to the prior distribution for the distance power parameter, $\beta$-Cauchy(0,1). The solid black line indicates the posterior distribution obtained. The vertical dashed red line indicates the maximum a posteriori estimate of $\beta$

**Population structure**. To test the existence of population genetic structure in anatomical space, we assessed the correlation between genetic (measured via $F_{ST}$ estimates) and geographical distance, using the Mantel test function in the adegenet R package[28] (Supplementary Fig. 1).

**Deconvolution of clonal populations**. Since the accuracy of the clonal deconvolution from mixed samples largely depends on the quality of the inferred VAFs, and copy-number variation is known to alter the allele frequency of somatic mutations in bulk tumor samples, somatic calls showing a VAF < 0.10, with a read depth <20 in all tumor and healthy samples, and/or overlapping with copy-number events were filtered out prior to clonal deconvolution. The number of tumor clones, as well as their genotype sequences, was then inferred using CloneFinder[13], which has been previously shown to outperform other methods in both simulated and empirical datasets. We required a minimum read count of 40 and a mutant read count of 6. The clone frequency cutoff was set to 0.075. The binary clonal sequences generated (i.e., A = reference status; T = alternative alleles) were then modified by changing the binary alleles into the corresponding nucleotide state. The consistency of the clonal genotypes inferred was evaluated by applying a different clonal deconvolution method, LICHeE[20], to the same dataset. LICHeE was run using the following thresholds: minVAFPresent = 0.075, maxVAFValid =

0.7, maxVAFAbsent = 0, minClusterSize = 2, minPrivateClusterSize = 1, max-ClusterDist = 0.1, outputTrees = 1 (Supplementary Fig. 3).

**Phylogenetic model fitting, reconstruction, and dating**. Bayesian phylogenetic analyses were performed using BEAST 2.4.7 (ref. [29]). First, the most appropriate evolutionary model (i.e., demographics and substitution rates) for our data was identified using Bayes factors[30]. A detailed description of the models tested can be found in Supplementary Table 1. For each candidate model, marginal likelihoods were obtained through a path-sampling analysis implemented in BEAST, using 100 independent Markov Chain Monte Carlo (MCMC) chains with 500,000 steps each. As a prior for the relaxed clock rate mean, a value of 4.6E-10 substitutions per site per generation derived experimentally for CRC[16] was used. For conversion to real time, a generation time of 4 days was assumed[16,31]. Moreover, since the clonal genotypes obtained only comprise variable genomic positions, an SNV ascertainment bias correction[32] was applied by modifying the "constantSiteWeights" attribute in the input XML file for BEAST. Posterior distributions under the model with highest support (i.e., Clock Model: Relaxed clock exponential; Tree: Coalescent Exponential Population) for the parameters of interest were obtained by running an MCMC chain during 100 million generations, sampled every 2000. Convergence was assessed using Tracer v1.6 (ref. [33]). After discarding the first 10%

of the samples as burn-in, point estimates for the different parameters were obtained using posterior means, and a maximum clade credibility topology was constructed using the median heights.

**Demographic analysis**. Demographic changes in the cancer cell population were inferred from a Bayesian skyline plot analysis carried out in BEAST 2.4.7. The same prior distributions described above were used, with the exception of the coalescent tree prior, which was set to "Coalescent Bayesian skyline". The final skyline reconstruction was obtained using Tracer v1.6, setting the number of bins to 100 and the age of the youngest tip to 0 (i.e., the time of collection looking backwards).

**Estimation of positive selection**. Somatic mutations were mapped on the BEAST tree using PAUP*[34] under maximum likelihood. Coding clonal sequences were obtained using the dndscv[35] R package, concatenated into a multiple sequence alignment, and analyzed using PAML 4.8a[36] to obtain maximum likelihood estimates of the non-synonymous/synonymous rate ratio (d$N$/d$S$) for the different branches of the inferred clonal genealogy in BEAST. The significance of these estimates was tested using likelihood ratio tests comparing a model assuming a single d$N$/d$S$ for the whole genealogy (model M0) and models assuming that a specific branch has a different d$N$/d$S$ than the rest (two-ratio model)[37].

**Inference of ancestral clonal ranges and migration history**. The ancestral spatial distribution of the clones was reconstructed using BayArea[6] upon the inferred BEAST genealogy, together with the observed geographic ranges of the tumor clones (i.e., presence/absence of each clone at each of the 16 sampled locations of the tumor) (Supplementary Fig. 4). Posterior distributions for the parameters of interest were obtained by running an MCMC chain during 100 million steps, sampling every 2000 generations. BayArea implements a probabilistic dispersal-extinction biogeographic model that considers how different lineages colonize new regions or disappear from them through time. To examine whether two-dimensional geographical distances played a role in the dispersal ability of tumor clones, two candidate biogeographic models were compared in BayArea using Bayes factors (computed with the Savage-Dickey density ratio method): the mutual-independence (null) model, in which clonal dispersal is not conditioned by spatial distance (i.e., distance power parameter, $\beta = 0$), versus a distance-dependent dispersal model, where the probability of dispersal is affected by spatial distance (i.e., $\beta > 0$: dispersal to nearby areas is more likely than to distant locations, or $\beta < 0$: long-distance dispersal events are favored over short-distance movements). In order to define the spatial distances, different 2D coordinate matrices describing the geographical location of the samples were explored (Supplementary Fig. 4). MACHINA[19] was run in parsimonious migration history mode (i.e., pmh_sankoff) using the inferred clonal genealogy from BEAST and setting the colon as the primary anatomical site (Supplementary Fig. 5).

**Reporting summary**. Further information on research design is available in the Nature Research Reporting Summary linked to this article.

## Data availability

Raw exome sequencing data have been deposited in the Sequence Read Archive database under the accession code PRJNA552658. All data supporting the findings of this study are available within the article and its supplementary information files. A reporting summary for this article is available as a Supplementary Information file.

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

## Acknowledgements

This work was supported by the European Research Council (ERC-617457- PHYLO-CANCER awarded to D.P.) and by the Spanish Ministry of Economy and Competitiveness—MINECO (BFU2015-63774-P awarded to D.P.). D.P. receives further support from Xunta de Galicia. J.M.A. is currently supported by an AXA Research Fund Post-doctoral Fellowship. J.M.C.-T. is supported by Grant PI15/01501-FEDER from the Instituto de Salud Carlos III, Ministry of Science, Innovation and Universities, Spain. We want to thank Diana Valverde for her help with the DNA extractions from several

samples. We want to additionally thank Nuria Estévez-Gómez, Pilar Alvariño and people from the Fundación Pública Galega de Medicina Xenómica (FPGMX) for their help with some of the experiments, and Tamara Prieto, Alberto Vicens, Harald Detering, Diego Mallo, and Sara Rocha for discussions. We also thank the Supercomputation Center of Galicia (CESGA) for providing computational resources.

## Author contributions

D.P. conceived and supervised the study. J.M.C.-T. obtained the tumor samples. S.P.-L. processed the samples. J.M.A. performed all the analyses. J.M.A. and D.P. wrote the manuscript with input from all other authors.

## Competing interests

The authors declare no competing interests.
