## [Peer Review File · Nature Communications]

Reviewers' comments:

Reviewer #1 (Remarks to the Author):

The manuscript by Alves et al applies a more traditional but several new biogeographic population genetics approaches such BayArea to reconstruct the somatic ancestry and spread of a single human colorectal cancer. These types of more sophisticated approaches are usually applied to macroscopic populations and the application to human cancer migrations is interesting and novel. They conclude that metastasis occurred very early, with initial spread to the liver and subsequent spread to hepatic and regional lymph nodes. Early metastasis has been inferred by others in many types of cancers, and this paper further reinforces this possibility. Although a single tumor, the study appears to be well-done and the analytical methods would be of interest to many investigators.

Several comments:

1) the subclone deconvolution is critical and there are many ways to find subclones from bulk sequencing data. It might be best to name the method (CloneFinder) upfront in the manuscript on page 2, line 43. The authors do note that another method (LICHeE) gave similar numbers of subclones and ancestries.

2) Another method (MACHINA) can also reconstruct migration histories of metastatic cancer cells. On page 4, line 117, it might be clearer to state more exactly what differences were found with MACHINA versus the current approach. MACHINA is illustrated in Fig S5, but perhaps a few more details are warranted.

3) Methods: an estimate of tumor purity would be helpful

4) Minor---Fig 2c: maybe add "population size" as well as N_e to the figure to improve clarity to cancer researchers.

Reviewer #2 (Remarks to the Author):

The authors analyze the clonal and migratory history of a metastatic colorectal cancer, showing that the metastases have monoclonal origin. Moreover, the authors study the temporal dynamics of this tumor. This is good work, and applying similar analyses to other metastatic tumors will improve our understanding of tumorigenesis and metastasis. Please see my comments below.

1. Systematic search for alternative phylogenetic trees.

The analysis is based on 21 tumor clones (+1 normal) inferred from VAFs of 475 SNVs in heterozygous diploid regions in 18 samples. Although the authors mention that alternative trees cannot be ruled out and that the inferred clonal genotypes are largely consistent with another method, I encourage the authors to perform a systematic search for alternative deconvolutions. This can be done similarly to Jamal-Hanjani et al. (NEJM 2017), e.g. first cluster SNVs according to VAFs followed by phylogenetic tree reconstruction using either CITUP or SPRUCE, methods that explicitly allow for alternative solutions. This would improve confidence in the conclusions drawn in the manuscript.

2. Migration analysis using MACHINA

The authors write:

"Also, our biogeographic approach allows for the presence of the same ancestral clone at more

than one location, and is able to consider the spatial distance among samples, unlike the approach of El-Kebir et al."

While the El-Kebir et al. approach does not weigh migrations according to spatial distance, it does support the presence of the same ancestral clone in multiple anatomical locations. I encourage the authors to run MACHINA on the phylogenetic tree inferred by BEAST. Tips that are present in multiple anatomical locations can be replaced by polytomies, with one leaf for each anatomical location. It would be interesting to see if the migration history described in the paper is reconstructed by MACHINA.

3. Timing

The authors use a prior of "4.6e-10 substitutions per site per generation" to determine timing. This number is very close to the mutation rate in normal somatic cells as described in two recent papers (Lynch, Trends in Genetics, 2010; Ju et al. Nature, 2017). I would be interested in understanding the effect of increasing the prior to larger rates on the timing. Moreover, it would be worthwhile to investigate if the tumor is microsatellite unstable.

Reviewer #3 (Remarks to the Author):

In this manuscript, the authors use standard tools from evolutionary biology to analyze cancer genomic data. Specifically, the authors consider the question of tracking the history, including the timing, of the development of cancer metastasis from an initial colorectal tumor. Using a variety of methods, the authors are able to infer the history of the spread of the cancer, and to provide dating information for the primary and metastatic tumors, measured in terms of years prior to clinical diagnosis. They also use the PAML software to infer branches along the inferred phylogenetic history that correspond to positive selection, and together with the mutations inferred to happen along these branches, this provides insight into the mutations involved in the metastatic spread of cancer.

The paper is well-written, and all analyses have been thoughtfully and carefully carried out. I have a few questions of clarification concerning the methodology — primarily things that I think should be clarified for readers who are not familiar with the tools typically used in organismal evolutionary studies. I also noticed a few small typos (listed below). All of my comments are minor; overall this paper represents an important contribution to the study of the evolutionary history of cancer within an individual patient. It is well-done and appropriate for publication in this journal.

Methodological questions:

(1) For the demographic analysis, it is mentioned (lines 201+) that coalescent tree priors were used. I don't understand why the coalescent should be involved in this case, since the cells reproduce clonally. This needs some justification. Likewise, for the BEAST analysis, only coalescent prior distributions were considered for trees (lines 194-195 and Supplementary Table 2).

(2) The phylogenetic tree inferred using Bayesian methods shows many nodes that are inferred with low posterior probability. Fortunately, the deeper nodes seem to be well-supported, which allows the subsequent analyses to proceed. However, I wondered about the appropriateness of the set of substitution models considered for cancer evolution. In particular, the standard models implemented in BEAST allow back-substitution, which is often considered uncommon for cancer clonal populations. The authors should comment on this.

(3) The test for selection in PAML are interesting. It seems that the tests are applied to the entire data set, which led me to wonder what would happen if only a subset of the genes were under

selection along a branch. How does the current application of the tests in PAML allow one to detect genes and/or gene-branch combinations that might be under strong selection? The authors should comment on exactly what types of selection they expect to detect with these tests.

(4) In the tree in Figure 2, mutations have been assigned to branches of the tree. How was this assignment done? Does this imply that *every* descendent branch has this mutation?

(5) How was sequencing error accounted for? Or is it assumed that following filtering, sequencing error is negligible? This should be commented on.

(6) Are the data available somewhere? I didn't see any location given where the data could be downloaded. Supplementary Table 1 is missing.

Minor typos:

— line 34, "Albeit the extensive intratumor heterogeneity" — this phrase doesn't make sense. Reword.

— line 53, in "pointing out to changes", remove the word "to"

— "Finally" is misspelled in the caption to Supplementary Figure 6

— Supplement: change "LICHeE was ran" to "LICHeE was run"

We are very grateful for the swift reviews and the constructive criticism from the three reviewers. Below we address all concerns and describe our revisions for this manuscript.

Reviewer #1 (Remarks to the Author)

The manuscript by Alves et al applies a more traditional but several new biogeographic population genetics approaches such BayArea to reconstruct the somatic ancestry and spread of a single human colorectal cancer. These types of more sophisticated approaches are usually applied to macroscopic populations and the application to human cancer migrations is interesting and novel. They conclude that metastasis occurred very early, with initial spread to the liver and subsequent spread to hepatic and regional lymph nodes. Early metastasis has been inferred by others in many types of cancers, and this paper further reinforces this possibility. Although a single tumor, the study appears to be well-done and the analytical methods would be of interest to many investigators.

AU.1.0: We want to thank the reviewer for the comments and the time taken to revise our manuscript.

Several comments:

1) the subclone deconvolution is critical and there are many ways to find subclones from bulk sequencing data. It might be best to name the method (CloneFinder) upfront in the manuscript on page 2, line 43. The authors do note that another method (LICHeE) gave similar numbers of subclones and ancestries.

AU.1.1: We now first mention CloneFinder on page 2, as suggested.

2) Another method (MACHINA) can also reconstruct migration histories of metastatic cancer cells. On page 4, line 117, it might be clearer to state more exactly what differences were found with MACHINA versus the current approach. MACHINA is illustrated in Fig S5, but perhaps a few more details are warranted.

AU.1.2: We now provide more details in the discussion when comparing the results obtained with BayArea and MACHINA (page 4, lines 125-130).

3) Methods: an estimate of tumor purity would be helpful

AU.1.3: We have estimated tumor purity and added this information to Supplementary Table 1.

4) Minor---Fig 2c: maybe add “population size” as well as N_e to the figure to improve clarity to cancer researchers.

AU.1.4: We have modified this label so now it reads “Historical tumor effective population size (N_e)”.

Reviewer #2 (Remarks to the Author):

The authors analyze the clonal and migratory history of a metastatic colorectal cancer, showing that the metastases have monoclonal origin. Moreover, the authors study the temporal dynamics of this tumor. This is good work, and applying similar analyses to other metastatic tumors will improve our understanding of tumorigenesis and metastasis. Please see my comments below.

AU.2.0: We want to thank the reviewer for the comments and the time taken to review our manuscript.

1. Systematic search for alternative phylogenetic trees.

The analysis is based on 21 tumor clones (+1 normal) inferred from VAFs of 475 SNVs in heterozygous diploid regions in 18 samples. Although the authors mention that alternative trees cannot be ruled out and that the inferred clonal genotypes are largely consistent with another method, I encourage the authors to perform a systematic search for alternative deconvolutions. This can be done similarly to Jamal-Hanjani *et al.* (NEJM 2017), e.g. first cluster SNVs according to VAFs followed by phylogenetic tree reconstruction using either CITUP or SPRUCE, methods that explicitly allow for alternative solutions. This would improve confidence in the conclusions drawn in the manuscript.

AU.2.1: Indeed, there are literally dozens of clonal deconvolution methods that we could have tried. As far as we know, the most comprehensive (in silico) benchmarking of clonal deconvolution approaches is that of Miura *et al.* (2018). Under the scenarios explored in that article, CloneFinder and LICHeE –the methods we used here– were the best ones.

Sayaka Miura, Karen Gomez, Oscar Murillo, Louise A Huuki, Tracy Vu, Tiffany Buturla, Sudhir Kumar, Predicting clone genotypes from tumor bulk sequencing of multiple samples, *Bioinformatics*, Volume 34, Issue 23, 01 December 2018, Pages 4017–4026, <https://doi.org/10.1093/bioinformatics/bty469>

Nevertheless, as suggested, we did our best to run CITUP and SPRUCE on our SNV dataset. Unfortunately, we were not able to run CITUP due to some problems related to the compilation. We tried solving these issues directly with the authors using the tool's bitbucket maintenance webpage, but unfortunately our questions went unanswered after 1 month (<https://bitbucket.org/dranew/citup/issues/17/error-running-citup-after-installing-with>).

We also had problems to run SPRUCE on our cluster, but in this case the developer (Mohammed El-Kebir) was very kind and offered himself for running the analysis. He sent us a “preliminary” clustering of all SNVs using a Gaussian mixture model on the variant allele frequencies, which resulted in 9 distinct mutation clusters. Afterwards, the enumerate function of SPRUCE (ran by the author and by ourselves, with the same result) to infer all possible alternative cluster trees and eight different solutions were obtained. Below we show these results:

Figure 1. Solution space inferred using the SPRUCE algorithm. Rectangles correspond to the different mutation clusters identified using a Gaussian mixture model on the VAFs of the 475 SNVs. Each edge is labeled by the number of trees in which it occurs.

Interestingly, although there is some overlap in the mutation clusters inferred with SPRUCE and CloneFinder (e.g., cluster_2, cluster_7 and cluster_11 correspond to clones Q, O and S detected by CloneFinder), some bizarre mutation clusters were also inferred by SPRUCE. In particular, cluster_4 from SPRUCE corresponds to a very large mutation cluster comprising the majority of mutations found in most metastatic sites sampled. However, since essentially all metastatic samples carry private mutations, it does not seem very realistic to assign all these mutations to a single, unique cluster/clone. Similarly, cluster_0 comprises mutations found in samples C2 and L4. However, and as we show in the table below, most of these mutations are private to either sample C2 or sample L4, suggesting that including them in a single cluster/clone is likely an artifact.

Table 1. Variant allele frequency at C2 and L4 of a subset of mutations comprising Cluster_0.

CHR	POS	C2	L4
chr12	56420857	0,00	0,15
chr14	69951885	0,20	0,00
chr16	67970187	0,30	0,00
chr19	13883034	0,34	0,00
chr19	20808338	0,13	0,00
chr19	46522458	0,00	0,27
chr1	118570917	0,26	0,00
chr1	169515684	0,45	0,00
chr1	169515705	0,44	0,00
chr1	172579042	0,19	0,00
chr1	180382613	0,06	0,00

Given these caveats, we believe that using the results from SPRUCE would not be a good idea. We want to reiterate that in the manuscript we used the two best performing tools in Miura *et al.* (2018), and that they largely agree on the clones inferred, despite using quite distinct algorithms

2. Migration analysis using MACHINA

The authors write:

"Also, our biogeographic approach allows for the presence of the same ancestral clone at more than one location, and is able to consider the spatial distance among samples, unlike the approach of El-Kebir *et al.*"

While the El-Kebir et al. approach does not weigh migrations according to spatial distance, it does support the presence of the same ancestral clone in multiple anatomical locations. I encourage the authors to run MACHINA on the phylogenetic tree inferred by BEAST. Tips that are present in multiple anatomical locations can be replaced by polytomies, with one leaf for each anatomical location. It would be interesting to see if the migration history described in the paper is reconstructed by MACHINA.

AU.2.2: We have to disagree with the reviewer regarding the capabilities of MACHINA. The states of the standard parsimony model implemented in MACHINA are *single locations*, and therefore by definition a node (=clone) in the tree can only be assigned to a single location at once. Indeed, different ancestral state reconstructions can be equally parsimonious, and in that case, uncertainty arises about the particular state at a given node, so multiple *exclusive* states are assigned, meaning such node could be in location x OR in location y OR in location z. In other words, ambiguity is not the same as multiplicity, in the state model of MACHINA multiple locations at internal nodes represents ignorance rather than inferred ranges. A tip representation with polytomies does not change this feature. On the contrary, in the BayArea model, the states are the *range of locations* occupied by a given node, so a node can naturally be in location x AND in location y AND in location z.

As requested, we performed a new MACHINA analysis upon the phylogenetic tree inferred with BEAST, setting each sampled location as a different anatomical site. Moreover, since we sampled eight primary tumor locations, all of them were tested in turn as potential primary anatomical sites. This resulted in a total of **30,924** migration histories, which is obviously a very large number. Looking solely at the results where the primary anatomical site was assumed to be C3 (i.e., the primary anatomical site inferred using BayArea), we obtained 18 maximum parsimony (MP) histories that imply 19 migrations, 16 co-migrations and 4 seeding sites (all other migration histories required a larger amount of migration and/or comigration events). One of the 18 inferred MP histories is *fairly* similar to the biogeographic history reconstructed with BayArea, although it suggests an early metastatic dissemination followed by a subsequent migration back to the primary tumor (L1 -> C1). Altogether, we believe these MACHINA results are very inconclusive. We now mention this analysis in the supplementary information.

3. Timing

The authors use a prior of "4.6e-10 substitutions per site per generation" to determine timing. This number is very close to the mutation rate in normal somatic cells as described in two recent papers (Lynch, Trends in Genetics, 2010; Ju et al. Nature, 2017). I would be interested in understanding the effect of increasing the prior to larger rates on the timing. Moreover, it would be worthwhile to investigate if the tumor is microsatellite instable.

AU.2.3: We used a mutation rate experimentally derived from hundreds of colorectal cancers (CRCs) by Jones *et al.* (2008), whom in fact obtained a similar estimate of mutation rates for normal and neoplastic colorectal epithelial cells. Indeed, this value is an expectation and it may differ between CRC patients (e.g., Hu *et al.*, 2019), but we believe is a sensible (experimentally derived for CRC) and conservative (not large) value for a prior.

Still, we ran BEAST estimates with a mean mutation rate to 9.2e-10 (i.e., twice higher than the previous rate). We observed a considerable effect in the absolute time estimates, which were halved (tMRCA age: 3.44-3.21 years, mMRCA age: 2.10 years)

suggesting a clear impact of the prior on the absolute dates. Importantly, the relative age of the metastatic ancestor does not change much (i.e., that the mMRCA and the tMRCA were close in time is still very much supported), and hence our main conclusions are still the same. We can consider that we are being conservative in the sense that larger mutation rates would imply even faster evolution. These caveats are now discussed in the main text (page 3, lines 91-98).

Regarding microsatellite instability, immunohistochemical staining confirmed that this tumor is microsatellite stable. We have added this information to the main text (page 2, line 27; page 5, lines 149-151).

Reviewer #3 (Remarks to the Author):

In this manuscript, the authors use standard tools from evolutionary biology to analyze cancer genomic data. Specifically, the authors consider the question of tracking the history, including the timing, of the development of cancer metastasis from an initial colorectal tumor. Using a variety of methods, the authors are able to infer the history of the spread of the cancer, and to provide dating information for the primary and metastatic tumors, measured in terms of years prior to clinical diagnosis. They also use the PAML software to infer branches along the inferred phylogenetic history that correspond to positive selection, and together with the mutations inferred to happen along these branches, this provides insight into the mutations involved in the metastatic spread of cancer.

The paper is well-written, and all analyses have been thoughtfully and carefully carried out. I have a few questions of clarification concerning the methodology — primarily things that I think should be clarified for readers who are not familiar with the tools typically used in organismal evolutionary studies. I also noticed a few small typos (listed below). All of my comments are minor; overall this paper represents an important contribution to the study of the evolutionary history of cancer within an individual patient. It is well-done and appropriate for publication in this journal.

AU.3.0: We really appreciate the comments and would like to thank the referee for the time taken to review our manuscript.

Methodological questions:

(1) For the demographic analysis, it is mentioned (lines 201+) that coalescent tree priors were used. I don't understand why the coalescent should be involved in this case, since the cells reproduce clonally. This needs some justification. Likewise, for the BEAST analysis, only coalescent prior distributions were considered for trees (lines 194-195 and Supplementary Table 2).

AU.3.1: The use of a coalescent prior is not related to the clonal or non-clonal nature of the data. In fact, the coalescent priors in BEAST, and in most software tools, always assume no recombination (i.e., that a single genealogy underlies all the loci). The use of a coalescent prior is instead related to (1) is most suitable for trees describing the relationships between individuals in the same population/species (see for example Bromham et al., 2018), (2) we are sampling from a much larger, growing population, as opposed to work with different species, when a Yule or Birth-Death prior might be also adequate, and (3) we are modeling how the population size changes through time, and

the pattern of branch lengths should be sensitive to these changes (i.e., we need to consider demography). In addition, there is a coalescent for cancer cells (Ohtsuki and Innan, 2017) which is identical to the standard coalescent with growth (Slatkin and Hudson, 1991) after a simple scaling (twice the cell division rate) related to the time units and the consideration of overlapping generations.

Indeed, the n-coalescent process has been shown to be a proper approximation to multiple reproductive models, not just to Wright-Fisher (Kingman 1982), including explicitly the genealogy of cells dividing in a binary fashion (Shierup and Wiuf, 2010)

Bromham L, Duchêne S, Hua X, Ritchie AM, Duchêne DA, Ho SYW. 2018. Bayesian molecular dating: opening up the black box. *Biol Rev Camb Philos Soc.* 93(2):1165-1191. doi: 10.1111/brv.12390.

Kingman, J. F. C.. 1982. On the genealogy of large populations. *J Appl Probab* 19A, 27–43. doi: 10.2307/3213548

Ohtsuki H, Innan H. 2017. Forward and backward evolutionary processes and allele frequency spectrum in a cancer cell population. *Theoretical Population Biology* 117: 43-50. doi:10.1016/j.tpb.2017.08.006.

Schierup MH, Wiuf C. 2010 "The coalescent in bacterial populations"., Robinson, D. Ashley Falush, Daniel Feil, Edward J. (editors). *Bacterial population genetics in infectious disease*. Chapter 1, United States: Wiley-Blackwell. pp 3-19. doi: 10.1002/9780470600122.ch1

Slatkin M, Hudson RR. 1991. Pairwise comparisons of mitochondrial DNA sequences in stable and exponentially growing populations. *Genetics* 129: 555-562.

(2) The phylogenetic tree inferred using Bayesian methods shows many nodes that are inferred with low posterior probability. Fortunately, the deeper nodes seem to be well-supported, which allows the subsequent analyses to proceed. However, I wondered about the appropriateness of the set of substitution models considered for cancer evolution. In particular, the standard models implemented in BEAST allow back-substitution, which is often considered uncommon for cancer clonal populations. The authors should comment on this.

AU.3.2: We did not observe multiple mutations at a single site in our dataset. It is true that the Markov substitution models implemented in BEAST allow for substitutions to occur multiple times at the same site (i.e., finite-sites models; FSM), but obviously they do not force them to occur. Indeed, the infinite-sites model (ISM) typically assumed in cancer, is just a special case of a FSM. While the reviewer might be worrying about overfitting, it is well-known in phylogenetics that overfitting a substitution is much less of a problem than underfitting (e.g., Arbiza et al. 2012; Abadi et al., 2019). And specifically, for this case, we have (unpublished) simulations showing that the use of a FSM vs. ISM model does not decrease phylogenetic accuracy when the data is generated under an ISM.

Arbiza L, Patricio M, Dopazo H, Posada D. 2011. Genome-wide heterogeneity of nucleotide substitution model-fit. *Genome Biology and Evolution* 3:896-908. doi: 10.1093/gbe/evr080

Shiran Abadi, Dana Azouri, Tal Pupko & Itay Mayrose. 2019. Model selection may not be a mandatory step for phylogeny reconstruction- Nature Communications 10: 934.doi: 10.1038/s41467-019-08822-w

(3) The test for selection in PAML are interesting. It seems that the tests are applied to the entire data set, which led me to wonder what would happen if only a subset of the genes were under selection along a branch. How does the current application of the tests in PAML allow one to detect genes and/or gene-branch combinations that might be under strong selection? The authors should comment on exactly what types of selection they expect to detect with these tests.

AU.3.3: It is a question of the available number of mutations. In our dataset we do not have enough mutations in any given gene to test for selection on a gene-by-gene basis which would indeed be possible with enough data (also to estimate dN/dS ratios for single sites with the so-called dN/dS site models; or for combinations of genes and branches). Our approach here was to identify branches with some evidence of overall positive selection across genes (i.e., with global dN/dS > 1), and then list the genes with non-synonymous mutations in these branches.

With the type of tests carried out we were trying to identify variable overall selective pressures (positive or negative) along the tree according to:

dN/dS = 1 indicates neutrality

dN/dS < 1 indicates purifying selection

dN/dS > 1 indicates balancing selection or directional selection (positive)

(4) In the tree in Figure 2, mutations have been assigned to branches of the tree. How was this assignment done? Does this imply that *every* descendent branch has this mutation?

AU.3.4: We mapped the mutations on the BEAST phylogeny using PAUP* (<https://paup.phylosolutions.com>) under maximum likelihood. Because in our data mutations never happened twice in the same site (i.e., no observable homoplasy) then all descendant lineages have the given mutation.

(5) How was sequencing error accounted for? Or is it assumed that following filtering, sequencing error is negligible? This should be commented on.

AU.3.5: Given the relatively stringent filter applied to our mutation calls (i.e., minimum coverage of 20X for both tumor and healthy samples, minimum variant allele frequency (VAF) of 0.05, and minimum nucleotide quality score of 20), we believe that sequencing error is negligible in our set. We now mention this in the text (page 6, lines 180-181).

(6) Are the data available somewhere? I didn't see any location given where the data could be downloaded. Supplementary Table 1 is missing.

AU.3.6: The raw mapped reads have been deposited in the Sequence Read Archive (SRA) under the following project: PRJNA552658. Supplementary Table 1 is now in a separate file (SupplementaryTable1.xlsx). We are convinced we sent this Table within the first submission, but maybe we made a mistake. In such case, we apologize.

(7) Minor typos:

— line 34, “Albeit the extensive intratumor heterogeneity” — this phrase doesn’t make sense. Reword.

— line 53, in “pointing out to changes”, remove the word “to”

— “Finally” is misspelled in the caption to Supplementary Figure 6

— Supplement: change “LICHeE was ran” to “LICHeE was run”

AU.3.7: We removed the unclear statement, and fixed the typos. Thank you.

REVIEWERS' COMMENTS:

Reviewer #1 (Remarks to the Author):

The author have satisfactorily addressed my concerns in the revised manuscript.

Reviewer #3 (Remarks to the Author):

The authors have done a very good job of responding to the initial reviews, and I have no further comments. I feel that the manuscript is ready for publication.

REVIEWERS' COMMENTS:

AU: We are very grateful for the swift reviews and positive feedback from the reviewers. We feel that the review process has substantially improved our paper.

Reviewer #1 (Remarks to the Author):

The author have satisfactorily addressed my concerns in the revised manuscript.

AU: We thank the reviewer for her/his time and useful comments.

Reviewer #3 (Remarks to the Author):

The authors have done a very good job of responding to the initial reviews, and I have no further comments. I feel that the manuscript is ready for publication.

AU: We thank the reviewer for her/his time and useful comments.